# Development of an Inertial Sensor Module for Categorizing Anomalous Kicks in Taekwondo and Monitoring the Level of Impact

**DOI:** 10.3390/s22072591

**Published:** 2022-03-28

**Authors:** Woo-Jin Jang, Ki-Kwang Lee, Won-Jae Lee, Si-Hyung Lim

**Affiliations:** 1Department of Mechanical Systems Engineering, Graduate School, Kookmin University, Seoul 02707, Korea; pkqoxmf@naver.com; 2Department of Sports and Health Rehabilitation, Kookmin University, Seoul 02707, Korea; kklee@kookmin.ac.kr; 3Department of Sports Industry and Leisure, Kookmin University, Seoul 02707, Korea; wjlee@kookmin.ac.kr; 4School of Mechanical Engineering, Kookmin University, Seoul 02707, Korea

**Keywords:** Taekwondo, IMU sensor, accelerometer, impact, anomalous kick, artificial intelligence

## Abstract

In this study, an inertial measurement unit (IMU) sensor module and software algorithm were developed to identify anomalous kicks that should not be given scores in Taekwondo competitions. The IMU sensor module was manufactured with dimensions of 3 cm × 3 cm × 1.5 cm and consists of a high-g sensor for high acceleration measurement, a 9-DOF sensor, and a Wi-Fi module for wireless communication. In the experiment, anomalous kicks and normal kicks were collected by the IMU sensor module, and an AI model was trained. The anomalous kick determination accuracy of the trained AI model was found to be 97.5%. In addition, in order to check whether the strength of a blow can be distinguished using the IMU sensor module, an impact test was performed with a pendulum under the same test conditions as the impact sensor installed in the impact test setup, and the correlation coefficient was 0.99. This study is expected to contribute to improving scoring reliability by suggesting the possibility of discriminating anomalous kicks, which were difficult to judge in Taekwondo competitions, through the analysis of Taekwondo kicks using inertial data and impulses.

## 1. Introduction

Taekwondo is a popular sport in Korea that is loved by many people. It has gained worldwide popularity and has been adopted as an Olympic sport. Taekwondo has been continuously developed as an Olympic sport in various studies. In particular, these developments focus on the fair judgment of the matches the player plays.

The protector and scoring system (PSS) currently used in the sport has a built-in pressure sensor and a proximity sensor, so when it is necessary to judge a blow during a match, it transmits data to help the referee make a more accurate decision. The data is transmitted to the terminal PC that will carry out judgment during the match.

In previous Taekwondo matches, there have been problems regarding misjudgment, such as allowing only kicking techniques in which the striking sound is clearly audible according to the sense of the referee when judging a kick. In this regard, C. Falco, I. Estevan, and M. Vieten tried to classify the reaction time, execution time, and total reaction time of kicks most commonly used in matches through measurement. As a result, players are known to prefer kicking techniques with efficient movements that enable them to respond quickly to the opponent’s attacks and counterattack the opponents quickly [1]. H. H. M. Kwok compared the combat strategies of Taekwondo medalists and non-medalists in Olympic Taekwondo competitions. Medalists are known to use quick attack kick techniques such as roundhouse kicks, cut-down kicks, and push kicks more often [2]. It was very difficult to secure objectivity in judgment in such an environment where fast kicks are frequently used.

However, after using the PSS, the pressure sensor inserted into the PSS responds to even the slightest contact of the players to solve this problem and increase the accuracy and objectivity of judgment.

As a result, the PSS was able to detect kicking techniques that could have been missed depending on the senses of the referee in the past. This has led to a change in the kicking techniques often used by athletes since the introduction of electronic protective equipment at the Olympic Games. According to U. Moenig, the forelimb kick technique is weaker than the hind leg kick. However, after the introduction of the PSS, players preferred the forelimb kick technique because there are many easier techniques that are judged to score by the electronic guard. It is also said that the electronic judgment system used in the PSS does not discern whether the players are using the correct technique, so they are using various anomalous kicks [3].

As a result, due to the sensitive reaction of the PSS, it is possible to score by just rubbing the body without an official kick technique, so the players focus on techniques that can score with minimal body movement even if it is not an official kick. In particular, the monkey kick is not an official kick technique, but it is known as a kick that can score efficiently and is often used in matches. Despite this circumstance, the PSS does not discriminate between anomalous kicks such as the monkey kick, so it cannot prevent these kicks from being used in matches.

Anomalous kicks, such as monkey kicks, are very similar to the official kick technique and cannot be accurately identified with the naked eye. Therefore, it is difficult to find a way to ban anomalous kicks even in official Taekwondo matches. In this regard, studies are being attempted to develop Taekwondo into an objective sport by applying various monitoring techniques.

Castaneda et al., developed a PSS that attaches an inertial sensor to the body of a Taekwondo protector to detect the typical kicking techniques used in Taekwondo [4]. J. de Goma et al., used an RGB-D sensor and skeletal data from Kinect to determine Taekwondo behavior with a preprocessing-focused approach [5]. Dharmayanti et al., analyzed the kinematic characteristics of a Taekwondo punch technique using a motion analysis camera and inertial sensor [6]. F.Liu digitized Taekwondo motions through motion data acquired using a motion capture device, measured the similarity of Taekwondo motions between instructors and trainees, and used them for skill training [7]. Ishac et al., performed a quantitative analysis by selecting the cross-kick as an ideal index with 3D image analysis to study the technical movement characteristics of Taekwondo [8]. 

As such, many researchers have tried to acquire accurate motion data using optical and non-optical methods in various sports activities, including Taekwondo [9,10,11].

The monkey kick, classified as a typical anomalous kick, has a slightly different movement in some joints than the roundhouse kick, which is a regular kick with similar motions. The difference in joint movement between the two kicks is shown in Figure 1. In this study, we focused on the IMU sensor to analyze and understand the joint movements used in these kicks. IMU sensors have been used in various studies to quantify movements that occur in sports. Jacob et al. used IMU sensors to measure elbow joint movements in badminton players [12]. G. Yu et al., proposed the optimal location to attach the IMU sensor among body parts by capturing the rotational movements of skiers using the sensor [13]. C. J. Ebner and R. D. Findling used IMU sensors to analyze suitable attachment positions for automatic tennis stroke recognition [14]. Do et al., attached an IMU sensor to the ankle of a pedestrian and used the measurement data to calculate roll, pitch, and yaw to detect and calculate stride length [15].

In this study, a method of analyzing posture data using an IMU sensor module, including an inertial sensor, was used as a method to accurately discern anomalous kicks that occur frequently in Taekwondo matches. The traditional method of analyzing posture data is by use of an optical-based method, such as recognizing a player’s foot movement through graphic capture using a camera. This optical method is the best way to identify an anomalous kick. Before starting this study, it was verified using a motion capture camera to verify the method of identifying an anomalous kick using an inertial sensor. However, although the motion capture camera can accurately capture the player’s motion, it is difficult for players to use it in Taekwondo matches because the player has to attach markers to various places on the body. As a motion capture method that can solve this problem, a motion analysis system uses a markerless method [16]. However, this markerless method requires at least eight cameras installed in the stadium. This creates a cost problem in official matches that require accurate judgment, and the match place can be severely limited. In addition, the markerless method identifies the player’s motion by analyzing the movement according to the joint coordinate values. This identification process includes a calculation process according to the coordinate values, so it takes some time to derive the result. Due to the nature of the Taekwondo match, it requires quick and accurate judgment, so it is difficult to use it in Taekwondo matches in terms of time. In addition, there is a problem in that in real matches, the referee may move in close contact with the player to see the player’s action accurately at the moment when a judgment is required, resulting in a blind spot that cannot be captured by the camera. For this reason, it is difficult to use the markerless method in Taekwondo matches. In this study, considering these factors, a method for quickly discriminating an anomalous kick using an inertial sensor was proposed to be applied to Taekwondo matches.

To collect data for the classification of anomalous kicks and detection of hitting, an inertia module that can measure and receive data from each joint of the wearer and software that can synthesize the collected data were developed in this study. The manufactured 9-DOF IMU sensor module was attached to each body part (spine, pelvis, and both ankles), and the data was measured wirelessly. Based on the measured data, anomalous kicks were classified by referring to the movement of each joint area. In addition, the level of impact was determined by referring to the acceleration value of the IMU sensor module, and the accuracy of kick determination was improved by providing data on whether a kick was taken. An experiment was conducted to find out the accuracy of the anomalous kick classification when using the random kick technique by learning the data collected from the sensor in an AI model designed in advance for the classification of anomalous kicks. In addition, after attaching the IMU sensor module to a pendulum, the relationship between the measured acceleration and the level of impact was confirmed by striking the fixed target to which the impact measuring sensor was attached for each elevation angle of the pendulum. As a result, the possibility of increasing the accuracy of judgment in actual Taekwondo competitions was confirmed in this study by classifying anomalous kicks that should not be scored.

## 2. Materials and Methods

For the IMU sensor module in this study, a 3 cm × 3 cm × 1.5 cm PCB was fabricated using a 9-DOF sensor and a 3-axis accelerometer called a high-g sensor. The 9-DOF sensor consists of a 3-axis accelerometer, 3-axis gyro sensor, and 3-axis geomagnetic sensor. The 3-axis acceleration sensor has a measurement range of ±16 g, the 3-axis gyro sensor has a measurement range of ±2000 deg/s, and the 3-axis magnetometer has a measurement range of±4800 uT. This IMU sensor module is used to measure the movement of each joint of the wearer. The sensor range of the 3-axis accelerometer included in the 9-DOF sensor is narrower than that of the high-g acceleration sensor used separately. Since the instantaneous acceleration of the kicks by the players exceeds the measurement limit of the 9-DOF sensor, the high-g acceleration sensor should be used to grasp the accurate time point of hitting by the kicks. The instantaneous acceleration of the kick used by the player is close to about 200 g. To measure it, an additional acceleration sensor was used, and the acceleration measurement range of this sensor is ±200 g. Hereafter, this sensor is generally referred to as a high-g sensor. In addition, to wirelessly receive data output from each sensor, a Wi-Fi wireless communication module capable of communicating at 2.4 GHz was used. The ESP 8266 module used in this study supports Wi-Fi communication in the 2.4 GHz frequency range and supports network protocols such as TCP/UDP. Each sensor connects to the Wi-Fi wireless communication module via I2C communication to transmit data to the terminal through UDP communication. A block diagram of the IMU sensor module is shown in Figure 2.

For the manufactured IMU sensor module, an individual zero-offset calibration was performed for the 4 sensors (3-axis accelerometer, 3-axis gyro sensor, 3-axis geomagnetic sensor, and 3-axis high-g sensor) built into the module. The calibrated IMU sensor module, which is connected to a wireless Wi-Fi router, transmits the sensor data to the terminal PC. The data is sent to the self-developed software based on Unity (a development tool for motion capture analysis in the sports field), which was developed to receive data from a module, display and store it on a terminal PC. The Unity-based self-developed software receives 50 raw data (9 variables) per second from the module through Wi-Fi and completes the processing of the data by time zone.

The three types of data output from the 9-DOF sensors are converted into roll, pitch, and yaw data using AHRS (Attitude Heading Reference System) by referring to the Madgwick algorithm from the data collected from the Unity-based self-developed software. Additionally, the AHRS algorithm in this study used a filter algorithm developed by Madgwick et al. [17]. The roll, pitch, and yaw data are used to judge the wearer’s overall joint movement.

The 3-axis acceleration data obtained from the high-g sensor determines if the kicking technique is actually correct. The experiment shown in Figure 3 was performed to confirm the correlation between the 3-axis acceleration data and the actual hitting of the kick technique. Since the sensor value of the high-g sensor is the acceleration value of each axis according to time, the following Equation (1) was used to obtain the magnitude of the acceleration at that time.
(1)$$a_{cal}\left(t\right)=\sqrt{{\left\{a_{x}\left(t\right)\right\}}^{2}+{\left\{a_{y}\left(t\right)\right\}}^{2}+{\left\{a_{z}\left(t\right)\right\}}^{2}}$$

As shown in Figure 3, an electronic protection device was worn on a fixed target to which the impulse measuring sensor was attached, and the IMU sensor module was attached to the front of the pendulum to strike. At this time, the acceleration value measured by the high-g sensor of the IMU sensor module and the impulse value measured by the impulse measuring sensor were compared. For the comparison of measured values, 20 hits were carried out per experiment using a hitting pendulum while changing the rising angle of the pendulum in five steps. Figure 4 shows a graph of the mean of the measurements for each PSS’s threshold level.

The protector worn on the fixed target in the experiment was the actual PSS from the Olympic Games, and the pendulum strike angle was selected based on the intensity value output from it. The intensity value is output from the PSS as a grade value without a unit, and in the experiment, the angle at which the same measured value as the preset intensity value is output when the strike with the pendulum was applied. The grade of hitting strength of the PSS was based on the minimum grade recognized when a male player kicks in the Olympics, and each section was raised by multiples of five from the standard to measure a total of five sections. The correlation index of high-g measurements and kick strength was calculated to be 0.99 according to Equation (2). Therefore, it was confirmed that the acceleration value measured by the high-g sensor is a variable that affects the strike strength of a kick.
(2)$$Correl\left(X,Y\right)=\frac{\sum \left(x-\bar{x}\right)\left(y-\bar{y}\right)}{\sqrt{\sum {\left(x-\bar{x}\right)}^{2}\sum {\left(y-\bar{y}\right)}^{2}}}\;$$

The variable *x* is the average acceleration value from the IMU sensor module according to the impact level of the PSS, and *y* is the average value of the impact amount from the impact sensor according to the impact level of the PSS. A value of 0.99 was obtained by calculating the correlation index according to Equation (2). With this, it was confirmed that the acceleration value measured by the high-g sensor was a variable affecting the strike strength of the kick.

In addition, the 3-axis acceleration data obtained from the high-g sensor was used to refer to the point in time when the kick occurred in the process of pre-processing the data before training the data to a pre-designed AI. In order to refer to the time at which the kick occurred, the moment with the greatest acceleration was first found. Since the instantaneous acceleration generated during a kick exceeds the measurement limit of the accelerometer built into the 9-DOF sensor, a high-g sensor was used for accurate measurement. Here, since the sensor value of the high-g sensor is an acceleration value of each axis according to time, the magnitude of acceleration according to time is the same as in Equation (1).

In order to understand the pre-processing conditions of the data to be used by the AI to learn kicks, IMU sensor modules were worn on each body part and data were collected using roundhouse kicks and monkey kicks executed once each. In this case, data from the IMU sensor module worn on the foot that performed the kick was used to determine the timing of the kick. The results of the experiment are shown in Figure 5, and with this graph, the timing of kick occurrence could be identified.

The data measurement experiment was conducted with six instructor-level Taekwondo players. Monkey kicks and roundhouse kicks were measured by attaching IMU sensors to the spine, pelvis, and ankle joints used in Taekwondo kicks. The roundhouse kick is the most-used technique in Taekwondo matches, and it was chosen because it has a similar movement to the monkey kick. Kicks were measured 10 times for each kick technique per person against a fixed target.

Figure 6 shows an example of wearing the IMU sensor module and the axial direction when the IMU sensor module is worn. In Taekwondo, there are many variables that must be considered for each kicking technique, such as different rotations of joints among players and changes in physical quantity due to differences in weight. Due to the occurrence of these variables, the judgment by referees varies even when the same kick is used. AI was used to solve these problems and to ensure the accuracy and objectivity of judgments. In the experiment in this paper, data were collected by attaching IMU sensor modules to various body parts of players and having the players move. The measured data were trained on a pre-designed AI model to check its accuracy. An artificial intelligence model was created using Keras, an artificial intelligence library, and the measurement data of each joint collected from a person with the IMU sensor module was combined into one sample for learning. As a result, six samples were trained for the AI model. In the process, the data collected through the IMU sensor module record the movement of each joint. Then, each joint data collected from the player is merged to make one sample. After learning about the monkey kick and roundhouse kick, an experiment was conducted to check the verification accuracy of the artificial intelligence model. In the experiment, participants wore IMU sensor modules and randomly made 10 roundhouse kicks and 10 monkey kicks. After measuring the data, it was checked whether the monkey kick, an anomalous kick technique, could be classified among the data collected from the IMU sensor module. The learning sample used in artificial intelligence was performed by merging data from IMU sensor modules collected from four body parts measured using kicks into one.

## 3. Results

### 3.1. AI Model Setup and Evaluation Method

In this section, data for each joint were learned using Keras from the kick data collected by the IMU sensor module in the experiment, and the output results are shown. There are several types of deep learning neural networks, but we used convolutional neural networks, and in this study, we used Keras’ sequential model. This model was used to easily train and test the data collected by the IMU sensor module through experiments. The developed AI model uses Adaptive Moment Estimation (ADAM) based on gradient descent as an optimizer [18]. For data learning of artificial intelligence, the data collected by the IMU sensor module were edited. As for the criterion for editing, the time when the maximum value was measured at the high-g sensor was selected to find the time when the kick occurred. The collision time was set based on the maximum value of the 3-axis acceleration of the 3-axis high-g sensor, $a_{cal}$. The artificial intelligence model used in this experiment was trained, as shown in Figure 7.

A confusion matrix was used to evaluate the classification performance of the trained AI model. The confusion matrix, shown in Table 1, is an indicator used to evaluate the performance of the classification model on a set of test data for which the actual value is known.

Here, TP is the number correctly predicted by the model for the actual roundhouse kick data, TN is the number correctly predicted by the model for the actual monkey kick data, FP is the number of roundhouse kicks predicted by the model for the real monkey kick data, and FN is the number predicted by the model as a monkey kick for the real roundhouse kick data. With this confusion matrix, we examined the performance indicators of this model, such as accuracy, precision, and recall, which are used as evaluation indicators for the correct answer output by the model. Each indicator is as follows: precision, also known as a positive predictive value (PPV), is the ratio of data predicted to be accurate in a model to actual accurate data. The precision is shown as in Equation (3).
(3)$$\mathrm{Precision}=\frac{\mathrm{TP}}{\mathrm{TP}+\mathrm{FP}}$$

Recall, also known as sensitivity or hit rate, is the proportion of data that the model predicts to be correct among the data that is actually correct, as shown in Equation (4).
(4)$$\mathrm{Recall}=\frac{\mathrm{TP}}{\mathrm{TP}+\mathrm{FN}}$$

Accuracy is the ratio of correctly predicted data among all data and is shown in Equation (5).
(5)$$\mathrm{Accuracy}=\frac{\mathrm{TP}+\mathrm{TN}}{\mathrm{TP}+\mathrm{TN}+\mathrm{FP}+\mathrm{FN}}$$

### 3.2. Evaluation Results

Using the pre-processed data, the learning convergence process of the AI model used in this experiment is shown as a graph in Figure 8. In the experiment for evaluating the learned artificial intelligence model, each experimenter randomly performed a roundhouse kick or a monkey kick 10 times, and a total of 40 kicks were performed from 4 experimenters. Table 2 shows the prediction results of the AI model for the data from the 40 kicks as a confusion matrix.

If the model evaluation index mentioned in Section 3.1 is used for the results predicted by the artificial intelligence in the experiment, the following Equations (6)–(8) are obtained. The result of calculating precision is shown in Equation (6).
(6)$$\mathrm{Precision}=\frac{20}{20+0}=1.00$$

The result of calculating recall is shown in Equation (7).
(7)$$\mathrm{Recall}=\frac{20}{20+1}=0.95$$

The result of calculating accuracy is shown in Equation (8).
(8)$$\mathrm{Accuracy}=\frac{20+19}{20+19+0+1}=0.975$$

## 4. Discussion

In this study, an inertial measurement unit (IMU) sensor module and software algorithm were developed to identify anomalous kicks that should not be given scores in Taekwondo competitions. The IMU sensor module was manufactured with dimensions of 3 cm × 3 cm × 1.5 cm and consists of a high-g sensor for high acceleration measurement, a 9-DOF sensor, and a Wi-Fi module for wireless communication. The manufactured IMU sensor module measures not only the wearer’s joint movement data but also the degree of impact when a blow occurs during a match. In the experiment to check the classification accuracy of anomalous kicks with the IMU sensor module, the most representative anomalous kick, or the monkey kick, and the official kick, or roundhouse kick, were tested. The artificial intelligence model trained on the collected data achieved 97.5% accuracy, 95% recall, and 100% precision for the random kick technique used by the player. In addition, in order to check whether the strength of a blow can be distinguished using the IMU sensor module, an impact test was performed using a pendulum under the same test conditions as the impact sensor installed in the test setup, and the correlation coefficient was 0.99.

Although a further study will be necessary to compare the anomalous kicks’ judging performance between optical method using cameras and non-optical method using IMU sensors, this study is expected to contribute to improving scoring reliability by suggesting the possibility of discriminating anomalous kicks, which were difficult to judge in Taekwondo competitions, through the analysis of Taekwondo kicks using inertial data and impulses.

## Figures and Tables

**Figure 1 sensors-22-02591-f001:**
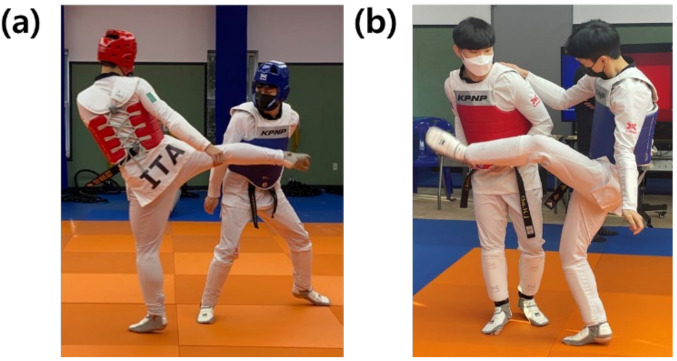
The difference in joint movement during kicks: (**a**) roundhouse kick, or the most representative anomalous kick and (**b**) monkey kick, or the official kick.

**Figure 2 sensors-22-02591-f002:**
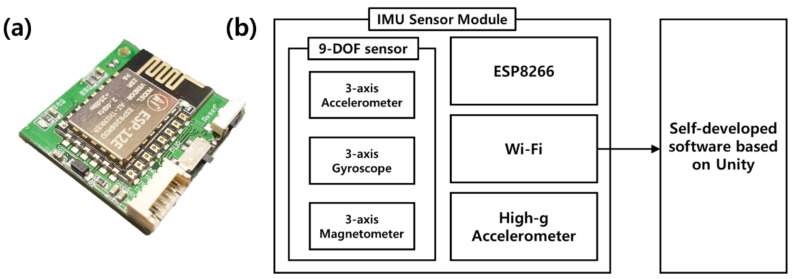
(**a**) Developed IMU sensor module and (**b**) block diagram of the IMU sensor module and self-developed software based on Unity.

**Figure 3 sensors-22-02591-f003:**
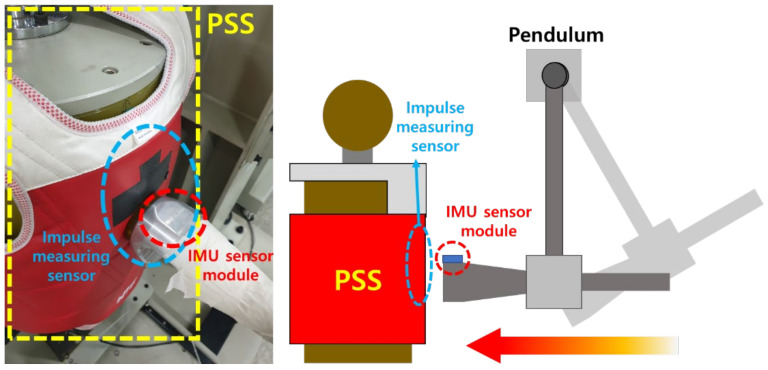
Experimental setup to check the correlation between the acceleration data from the IMU sensor module and the impulse measuring sensor attached to the electronic protection device, or PSS.

**Figure 4 sensors-22-02591-f004:**
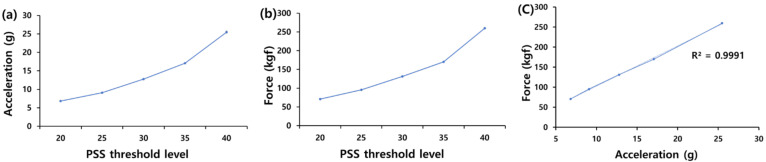
(**a**) Average acceleration data for each PSS’s threshold measured by high-g sensor. (**b**) Average force data for each PSS’s threshold measured by impulse measuring sensor. (**c**) Linear fitting graph of average acceleration vs. average force.

**Figure 5 sensors-22-02591-f005:**
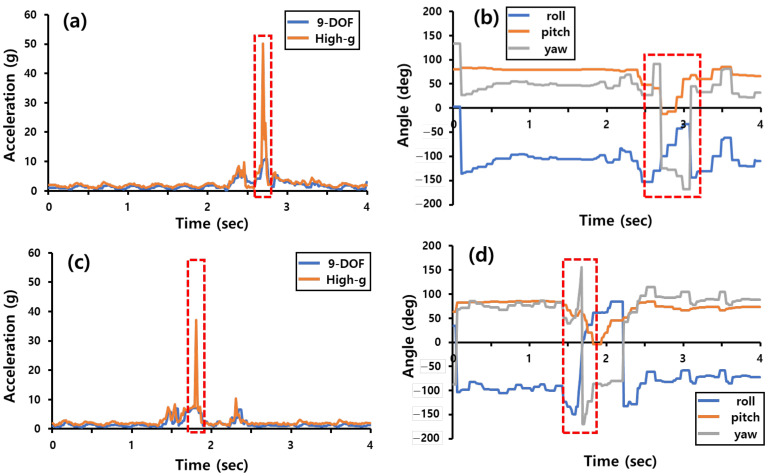
Acceleration and roll/pitch/yaw angle is obtained from the IMU sensor module when a roundhouse kick and a monkey kick are performed. (**a**) Acceleration data vs. time during one roundhouse kick, (**b**) roll/pitch/yaw angle vs. time during a roundhouse, (**c**) acceleration data vs. time during one monkey kick, (**d**) roll/pitch/yaw angle vs. time during one monkey kick.

**Figure 6 sensors-22-02591-f006:**
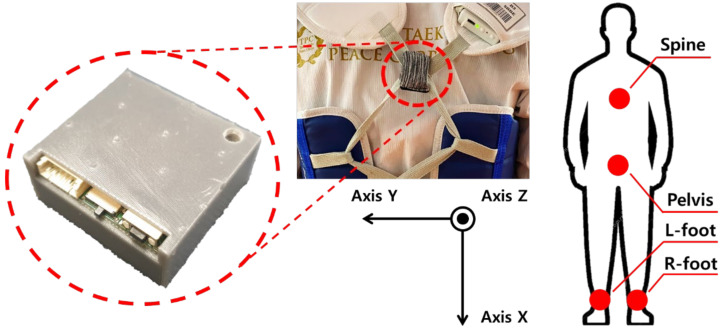
Installation of the IMU sensor modules on a player and the axial direction when wearing the IMU sensor module.

**Figure 7 sensors-22-02591-f007:**
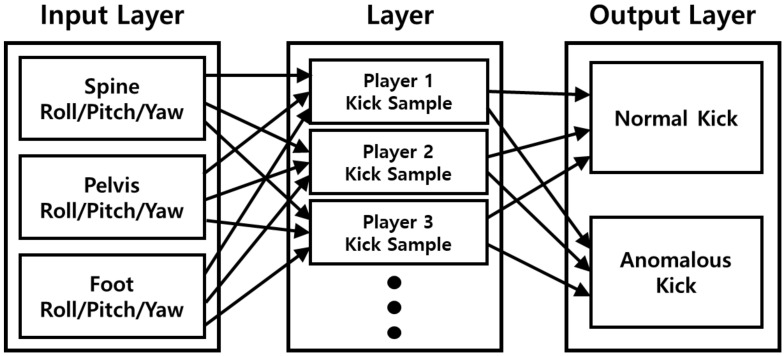
Block diagram of the designed artificial intelligence model.

**Figure 8 sensors-22-02591-f008:**
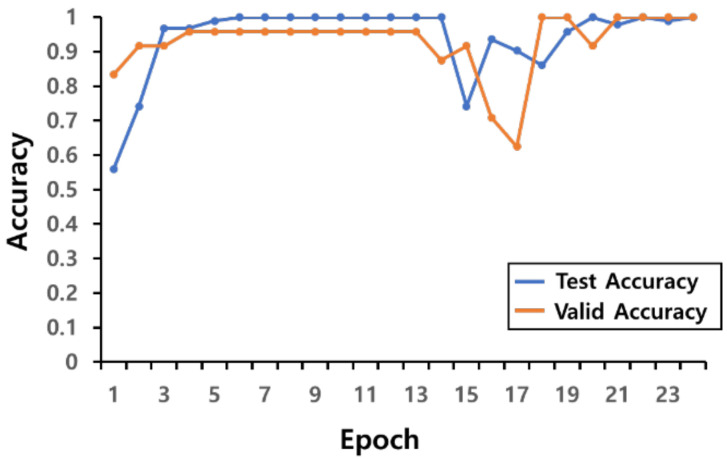
Learning convergence graph of AI model.

**Table 1 sensors-22-02591-t001:** Confusion matrix.

		Predicted	
		Positive	Negative
Actual	Positive	TP	FP
	Negative	FN	TN

**Table 2 sensors-22-02591-t002:** Confusion matrix for kick classification results.

		Predicted	
		Roundhouse Kick	Monkey Kick
Actual	Roundhouse kick	20	0
	Monkey kick	1	19

## Data Availability

Please contact the corresponding author for data requests.

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
