# Peer review of "Development of an Inertial Sensor Module for Categorizing Anomalous Kicks in Taekwondo and Monitoring the Level of Impact"

_sensors, 2022, doi:10.3390/s22072591_

Round 1

Reviewer 1 Report

interesting work, but it could be better (and also necessary) to compare the wearable approach with the optical-based method. please improve.

Author Response

Thank you very much for reviewing my manuscript.

Your question has been answered in the attached open review file. Also, the manuscript has been revised, so please see the attachment.

Reviewer 2 Report

You mention in your study that you are using a high-g sensor but do not give technical details. It is not possible for other scientists to understand what you did. You further mention a Wi-Fi module but from figure 1 I can see that you are using an ESP-12E which in a micro controller with built in Wi-Fi. SO it is unclear for me if you are doing some kind of processing in the micro?
You do not provide any sensor range (how much g, how much degrees per secodn, and so on).

Questions related to the manuscript:

line 16: I guess you mean 3 x 3 x 1.5 cm and not cm3

line 17: What does high-g mean? 50g, 100g?

line 103-107: is repeating information and can be deleted

line 120: same again I guess you mean cm and not cm3

line 128: You are using a ESP microcontroller with built in Wi-Fi and no Wi-Fi module. Please also provide the technical specifications for the used ESP.

line 132: pleas update figure 1 accordingly

line 135-136: what are the sensor ranges? Please also provide technical details of the used sensors (manufacturer, model, and so son)

line 139: What software is that (pre-developed software)? Please provide a reference.

line 141-142: repeating information

line 143: I guess you mean it is recorded at 50Hz and not transmitted via Wi-Fi with 50Hz. Please correct.

line 145: ...data using AHRS... Please provide a reference

equation 1: I guess you mean acal(t) =sqrt(ax(t)2+ay(t)2+az(t)2).
so at any point t

line 161: 20 blows.... is that correct terminology or is 20 kicks better?

line 193: ...the foot that used the kick.... may be better: ...the foot that performed the kick....

line 201: imu should be written IMU

line 229: ...uses Adaptive Moment Estimation (Adam)... please provide reference

line 278: again cm3 I guess you mean cm

General:

  • in your results section you mention a correlation coefficient but do not show a graphical representation. It would be nice for the reader to see a graphical one.
  • if you are comparing two different methods you should investigate doing a statistical analysis using Bland-Altman

Author Response

(The authors gave the same response as above.)

Reviewer 3 Report

Please find my comments in the attached file. This is an interesting paper but the questions must be address regarding the need for using AI platforms etc. 

Author Response

(The authors gave the same response as above.)

Reviewer 4 Report

Paper describing developed sensor system for taekwondo. Introduction should be extended in means to add some illustrations of types of Taekwondo kicks. 

Experiment should be conducted with at least 3 different persons performing kicks to prove AI generalization and make statistics valid!

Claiming 100% precision on 40 test samples and one experimenter is not very convincing. Extension of experiment must be conducted!

Some further comments:

L15: Not clear enough for persons not familiar Taekwondo. Is just anomalous kick not scored or scoring is totally stopped?

L17: What kind of 9DOF sensor? Not clear enough.

L29: In Introduction add some illustrations of types of Taekwondo kicks, especially legit and anomalous kicks when explaining them in text.

L43: Rephrase. What does "respond quickly to individuals" mean?

L68: Rephrase: "As such . ..,  such as ... "sounds weird

L91: CAPS LOCK “imu”

L92: CAPS LOCK “imu”

L101: Same as previous paragraph! Almost no new information is given!

L123: „The PCB“ -> „IMU sensor module“ - PCB itself cannot measure anything, it is a part of IMU sensor module

L125: Rephrase, doesn't sound right, not clear enough.

L133: schematic diagram -> block diagram

L135/136: Sentence is not easily understandable. Rephrase.

L138: What is "iptime"? Manufacturer? If so - it is better to write "wireless router".

L145: Madwick -> Madgwick

L165: Add scatter plot in order to better illustrate statistic properties of measurements. Keep mean value points marked.

Equation 2: What are x, y? Label and denote variables. Get better resolution of picture with formula!

L178: First part of sentence gives no info about variables but it gives value of correlation. It would sound better if rephrased. Example: "Correlation index of high-g measurements and kick strength is calculated to be 0.99 according to Eq. 2."

L228-L232: Unnecessary formulae, not vital in any sense for this paper.

L233: Second part of sentence is not clear. Rephrase.

L236: Write few paragraphs about type and structure of model.  What is it based on? Feed-forward neural networks? Figure 6 is too technical, make it more abstract and illustrative.

L239: Figure 6 is too technical, make it more abstract and illustrative.

L253: ration -> ratio

L263: Experiment should be conducted with at least 3 different persons performing kicks to prove generalization and make statistics valid!

Claiming 100% precision on 40 test samples and one experimenter is not very convincing.

Author Response

(The authors gave the same response as above.)

Round 2

Reviewer 1 Report

The authors did not response directly to the previous comment, which is, to compare the method with a vision-based approach. A major revision is still suggested. 

Author Response

(The authors gave the same response as above.)

Reviewer 3 Report

Thank you for addressing my previous comments. I do not have further suggestions for this manuscript. 

Author Response

(The authors gave the same response as above.)

Reviewer 4 Report

Revised paper can be accepted.

Author Response

(The authors gave the same response as above.)

Round 3

Reviewer 1 Report

The authors did not answer the previous comments. A major revision is still recommnended from this reviewer.

Author Response

Dear reviewer

Thank you for your valuable comment.

Your question has been answered in the attached open review file. Also, the manuscript has been revised, so please see the attachment.
